# Large-Scale Meta-Longitudinal Microbiome Data with a Known Batch Factor

**DOI:** 10.3390/genes13030392

**Published:** 2022-02-22

**Authors:** Vera-Khlara S. Oh, Robert W. Li

**Affiliations:** 1United States Department of Agriculture, Agricultural Research Service, Animal Genomics and Improvement Laboratory, Beltsville, MD 20705, USA; 2Department of Data Science, College of Natural Sciences, Jeju National University, Jeju City 690-756, Korea

**Keywords:** meta-longitudinal microbiome data, batch factor, longitudinal differential abundance test, functional enrichment

## Abstract

Data contamination in meta-approaches where multiple biological samples are combined considerably affects the results of subsequent downstream analyses, such as differential abundance tests comparing multiple groups at a fixed time point. Little has been thoroughly investigated regarding the impact of the lurking variable of various batch sources, such as different days or different laboratories, in more complicated time series experimental designs, for instance, repeatedly measured longitudinal data and metadata. We highlight that the influence of batch factors is significant on subsequent downstream analyses, including longitudinal differential abundance tests, by performing a case study of microbiome time course data with two treatment groups and a simulation study of mimic microbiome longitudinal counts.

## 1. Introduction

The microbiome, referred to as “the entire habitat, including the microorganisms (bacteria, archaea, lower and higher eukaryotes, and viruses), their genomes (i.e., genes), and the surrounding environmental conditions” [1], plays an important role in the host physiology, nutrition and development. The gut microbiome modulates host immunity [2] and protects the host against invading pathogens [3]. Recently, the microbiome has been linked to various diseases, such as ageing-related diseases [4], autism [5], cancer [6,7], gastrointestinal disorders [8,9], obesity and metabolic disorders [10,11]. The gut microbiome has recently been touted as a therapeutic target or as a drug target [12].

Gut microbial communities are inherently dynamic. Gut microbial composition and interactions fluctuate temporally or in response to numerous biotic and abiotic factors or perturbations. There are two unique features of the longitudinal (time-series) study design—i.e., the fact that time imposes an inherent, irreversible ordering on samples; and the fact that samples exhibit statistical dependencies that are a function of time—that make this the ideal method to understand the structure and function of the microbiome. Longitudinal and repeated sampling allow us a better and quantitative understanding of intraindividual variation. As a result, an increasing number of microbiome studies are taking advantage of the longitudinal design [13,14,15,16,17,18], partially driven by the drastic reduction in sequencing cost in the past few years. However, the longitudinal design often suffers from irregular sampling intervals and missing data. Proper methods for ad hoc adjustments to account for the missing data are needed. In addition, there is currently a lack of generic frameworks that enable unified statistical inferences of time series data obtained across multiple research centres via different platforms and from different phenotype measurements (multi-omics). In this study, we fully discuss the critical batch issues [17,19,20,21,22,23,24,25] that have emerged in the microbial community in the comprehensive analyses of both large-scale integrated longitudinal microbiome data and simulation studies.

## 2. Results

### 2.1. Initial Inspection of Batch Factor and Whether It Is Statistically Significant

Our main experimental design is based on the longitudinal time series data with two different groups at each time point in which an individual animal subject is repeatedly measured during the particular time period (from baseline Day 0 to Day 14 in trial I and further measured E0 and E2 at later dates in trial II). Moreover, there are control and target drug treatment group samples at each time point, resulting in 70 samples in trial I and 20 samples in trial II, respectively. Thus, there has been achieved the integrated longitudinal data with 90 samples in terms of two different trials with two different primer-sets. This large-scale meta-longitudinal microbiome data, where samples are integrated with two different trials (trial 1 versus 2), sequenced by distinct primer-sets (V3/V4 versus V1/V3) (Also see Figure 1 and Appendix A for the detailed sample profiling information), have been explored to determine whether the known batch factor of primer-sets is statistically significant by using the exploratory tools of guided PCA (principle component analyses) [23]. These metadata will be literally referred to as “uncorrected data” hereafter (see Figure 1A,B for a schematic illustration of both real data application and simulation schemes in this study). We examined whether there were any significant systematic biases due to the integrated samples of metadata by different days with different primer sets prior to detection of time-varying group differences in longitudinal differential abundance tests (Movie 1-(D)). Note that the known batch factor on different days has unbalanced sample sizes, and there are only two time points (E0 and E2, Days 22 and 50) with 20 samples in the second trial. The given delta value is defined by the proportion of the variance explained by the known batch factor, and it is computed by the ratio value of the proportion of total variance from the first component on guided PCA on numerator divided by that of unguided PCA on denominator, respectively. The corresponding statistical significance is calculated by the permutation procedures by randomly shuffling the labels of batch samples with 1000 times as the default in the package [23]. Based on the inspection with the quantified metrics, it is statistically nonsignificant (*p* value = 0.142) with a moderate delta value of 0.446 in the range of 0 to 1 [23]. The possibility is that there are unbalanced samples for a known batch factor, and these unbalanced samples are confounded with the main factor of time. Another possible reason could be that the variation by main factors to represent any significant difference for intergroups and time-varying trajectory patterns within each group in the upper panels is trivial when compared to other nuisance factors. Hence, we further aimed to examine the influence of the known batch factor in terms of biological impact on subsequent downstream analyses in depth by using uncorrected data, three types of batch corrected data, and marginal data. A total of five different types of batch handling procedures (i.e., uncorrected raw data with a known batch factor VERSUS three corrected data via batch identification tools (Also see online methods (1) Batch-correction methods) VERSUS marginal data in which the known batch factor is ignored by filtering out samples in trial II) were investigated in longitudinal differential abundance tests, clustering samples, sample classification, and functional enrichment pathway analyses (see Figure 1A).

### 2.2. Biological Impact of Batch Factor on Longitudinal Differential Abundance Test

A total of two different trials (trial 1 versus 2) combined with primer-sets were statistically tested to identify the set of candidate features (OTU_ID) representing temporally differential abundance (TDA) based on (1) uncorrected data, (2) three types of corrected data controlling batch factor, and (3) marginal data with trial 1 ignoring the batch factor. Note if a given feature is statistically significant in terms of group difference over time by a method at adjusted BH *p*-value = 0.05 after multiple correction, then it is called a TDA, otherwise, it is called TEA (temporally equal abundance) from a statistical test. It is similar to temporally differential expression (TDE) or temporally equal expression (TEE) in transcriptome data such as RNA-Seq. From each detection method, candidate lists at taxa levels selected by the multiple-corrected *p* values with an arbitrary threshold of 0.05 were compared. Based on the core intersection set indicating “always TDA calls” in four different detection methods in longitudinal differential abundance tests for each batch tool and its complementary set, Harman-corrected data show the batch removed data after correction. The other two different types of corrected data still showed the batch contaminated data on 3D-PCA in Movies (Movie 1-(A) to (E)). Note that the splinectomeR and dream methods showed no significant feature sets for the evaluated datasets. Accordingly, “always TDA calls” are defined as the intersection set between advanced tools, metaSplines, and metamicrobiomeR.

In the heatmaps with the core intersection sets for five different types of batch handling procedures, from Figure 2, as there exists some lowly abundant taxonomy levels, after filtering out those features via the default options in MicrobiomeAnalyst, the Harman batch correction method has better performance by demonstrating more explicit discrimination of two groups over time—i.e., at especially moderately or highly abundant taxonomy levels in Figure 3. In addition, the heatmaps by the list of complementary sets from other two correction tools, ARSyNseq and ComBatSeq, as well as uncorrected data, still showed the batch effect (see the samples in blue bar in Figure 4), whereas Harman-corrected data more clearly showed the batch removal pattern in Figure 4. Furthermore, based on the results from Figure 5A, Harman-corrected data present much tighter grouping clusters in intra-samples except in 3008 at D-14 and 3443 at E-2 subject (highlighted in stars) in control group on pvclust with complete [26], compared to others with more mixed-up patterns between two treatment groups to cluster samples (Figure 5B for uncorrected data). Dendrograms of plot trees after trimming the features with low abundance levels as default criteria in MicrobiomeAnalyst [27,28] also confirmed that Harman-corrected data showed a clearer pattern in sample clusters within a group, whereas uncorrected data failed to group more similar intra-samples within each group together (Figure 5C,D).

In fact, there are no biomarker gold-standard lists as the reference sets to define whether a particularly detected TDA or TEA (temporally equal abundance between groups over time) is biologically true in real data applications. Therefore, we further evaluated the biological impact of batch factors on functional enrichment pathway analyses by using MicrobiomeAnalyst [27,28], which incorporates α/β diversity, discrimination methods for feature selection, and biomarker analyses, including PICRUSt [29], a versatile web-based interface method for microbiome data in the following sections.

### 2.3. Biological Impact of Batch Factor on Functional Profiling Data Analyses

We further examined how the impact of batch source affects the functional enrichment analyses [27,28,29] based on the core intersection set of temporally differential abundance lists presented. First, abundance profiling at the taxonomic genus level is shown from core intersection sets detected by four different longitudinal differential abundance test methods in five different types of batch handling procedures for evaluation, showing the lesser extent of hierarchy of taxonomy genus levels on uncorrected data compared to others in bar graphs (Figure 5F), suggesting that the sophisticated batch removal should be considered prior to functional profiling data analyses. Next, β diversity profiling 2D-PCoA ensured the better performance of Harman-corrected data than others in inter- and intra-samples for two different groups, control versus treatment, as shown in Figure 5G,H, and Appendix A–C. Last, the results of overall performance in random forest sample classification consistently demonstrate smaller error rates on Harman-corrected data and marginal data, followed by ARSyNseq, ComBatSeq, and uncorrected data (Figure 5I,J for Harman and uncorrected data and see also Appendix A–F for the comparison). For the classification performance in samples of the control group, uncorrected and ComBatSeq-corrected data showed poor performance.

### 2.4. Biological Impact of Batch Factors on Network Modules of Disease-Associated Functional Enrichment Pathways

To examine how selected temporally differential abundance lists are enriched in a certain set of pathways for each of five different types of batch handling procedures, the corresponding results mapped by Kyoto Encyclopedia of Genes and Genomes (KEGG) orthology (KO) IDs generated from PICRUSt for the prediction of functional enrichment pathways were compared [27,28,29]. The significantly enriched networks of pathways selected by adjusted *p* value of 0.05 are given in Appendix A from the five different core intersection sets for evaluation. The top significantly enriched pathways are highlighted in blue in Appendix A. The Harman-corrected data showed the lowest number of significantly enriched pathways by picking up folate biosynthesis and styrene degradation as the top candidates when compared to others. The possibility is due to trimming the features with negative values on Harman-corrected data. Folate biosynthesis is the consensus of the top significantly enriched pathways from five different types of batch handling procedures. Thus, Harman-corrected data showed the least sensitivity detection rates as it has filtered out some features with negative abundant values on corrected data, which might be either zero or low abundant features in the original raw data and those are not meaningful features. For the comparison, the network enrichment KEGG pathways in overall functional profiling in association analyses are given in Figure 6A–E for five different types of batch handling procedures.

Furthermore, the selected core intersection sets were used for taxonomy-set enrichment analyses (TSEAs) [27,28] curated with host intrinsic and extrinsic environmental factors to identify how temporally differentially abundant lists are presented in interacting network modules of genetic variations, disease associations, and other types of external stimulated factors, such as drug treatments. Statistically significant enriched pathways are listed in Appendix A for three types of corrected data, uncorrected data, and marginal data. From the results, Harman-corrected data identified the top eight significantly enriched disease-associated terms from curated host-intrinsic taxonomy sets, including colorectal carcinoma, resistance to immune checkpoint inhibitors, and bacterial vaginosis. The commonly detected terms are highlighted in purple. In addition, the corresponding figure for the network modules of TSEAs defined by host intrinsic environmental factor sets is given in Figure 7A–E.

Collectively, this process ensures that the known batch factor substantially affects the results of the downstream analysis of identification of TDA OTUs in the longitudinal differential abundance test by presenting the behaviour of inconsistent detection of TDA from five different types of batch handling procedures. Similarly, this process exhibits the behaviour of discrepant results on the sample clustering analyses relying upon the different types of batch-corrected data. The results of random forest sample classification, functional diversity profiling data analyses, and functional enrichment pathway analyses also confirm that the biological impact of batch factors is not negligible and that batch-corrected data should be utilized in major statistical testing procedures. Overall, the Harman-corrected data showed more reliable results on all subsequent downstream analyses than the other two types of corrected (ARSyNSeq and ComBatSeq), uncorrected, and marginal data in real data application.

### 2.5. Performance of Batch Identification and Removal Methods When There Are Either Balanced or Unbalanced Sample Sizes for a Known Batch Factor in Simulations

Truly temporally differentially abundant features with L_down hockey stick-type trends and temporally equal abundant features have been generated via the gen_norm_microbiome function in the microbiomeDASim R simulator package [25,30]. Based on gold-standard lists, the performance for each batch correction tool on the identification of time-varying group differences in longitudinal trajectory tests by thoroughly comparing the estimated adjusted BH *p* values [31] from multiple dynamic methods has been evaluated by the area under the curve (AUC) comparing the true positives and true negatives [32].

From the AUCs in Figure 8, except for the datasets with the smaller number of individuals, overall, the Harman batch-correction method showed the best performance when compared to other batch correction methods, marginal, and uncorrected data. In particular, in terms of method-specific behavior, from the metaSplines and dream method, Harman still showed the best performance, followed by ComBatSeq with comparable performance. The splinectomeR method is too conservative, resulting in no significant features for all simulation scenario sets, confirming the results shown in real data applications. Thus, importantly, the advanced model controlling other significant confounding variables in this method does not work properly for contaminated data with various batch sources.

## 3. Discussion

More recently, two studies in transcriptome data in arrays were examined [22,33] to explore systematic unwanted batch sources against the main biological factors of interest.

In this study, we demonstrated how different patterns due to the known batch factor induced by the different trials in meta-longitudinal microbiome data are shown in the results of detection of statistically significant TDA OTUs and other downstream analyses. This demonstration has led to a more recently developed batch detection tool, Harman-corrected data, which presents better performance compared to other types of batch handling procedures based on heatmaps of core intersection set and its complementary set by demonstrating the clearer discrimination between two distinct groups over time, especially for the features at moderately or highly taxonomic levels, even though advanced microbiome-specific detection tools such as metaSplines and metamicrobiomeR [30,34] for longitudinal differential abundance test methods are employed. Sample clustering via pvclust and plot tree dendrograms also showed that intra-samples within each group in Harman-corrected data are more likely to be closely grouped together than uncorrected raw dat. Again, RF results confirm the lowest error rates in overall classification performance as well as each group for Harman-corrected data. β diversity analyses via PCoA also presented the better results in Harman-correction. In the functional enrichment analyses, Harman-corrected data showed the least sensitivity on the detection rates when compared to other types of batch handling procedures. The possible clues are that functional enrichment results largely depend on the size of a given input data (# of features) and Harman-corrected data have filtered out some features with negative abundant values, which might be either zero or low abundant features in the original data set and those are not meaningful features.

In simulations, Harman-corrected data outperformed other types of batch handling procedures in terms of AUCs [32] in various simulation scenarios.

Conclusively, common normalization procedures such as the global scaling method do not perfectly adjust unwanted systematic artefacts [17,19,20,22,24,25,28,34,35,36,37,38], and it is important to examine batch sources by using exploratory tools in a visualization manner and quantify metrics of how proportional data contamination induced by various lurking variables, including both known and unidentifiable factors, is observed. The influence of the biological impact of batch factors against the main factors was significant in the subsequent downstream analyses, such as longitudinal differential abundance tests and functional enrichment pathway analyses, as shown in our study, although metamicrobiomeR [34] and dream [39] to account for nuisance factors in the models were utilized. Thanks to the reduced cost of sequencing, it becomes feasible to conduct more complicated experimental designs, longer series of longitudinal data, larger-scale cross-sectional studies, and integrated metadata in different studies, trials, days, and laboratories [17,19,27,28,34,38], implying that various batch factors will be more universal phenomena in years to come. Taken together, it is necessary to develop unified approaches [17,40] instead of two-step approaches to integrate batch identification and its direct incorporation procedure in longitudinal differential abundance models, which are validated in large-scale comparative studies evaluating ranks with other existing methods in terms of true discovery rates. Thus, lastly, there are currently no comparative studies to evaluate and rank the performances with the aspects of longitudinal differential abundance tests as well as other down-stream analyses in terms of true discovery rates and power of detections by using the multiple longitudinal time series microbiome data or its meta data and simulations studies when there are identifiable or unidentifiable batch factors. As such, we could not include all of the commonly used longitudinal differential abundance test and batch methods in the microbial community for our current study. We are currently preparing for another submission of the next study to explore the large-scale of comparative study and to propose a novel unified Bayesian approach by including the popular and more recently developed methods such as ANCOM and MMUPHin [17,18]. Collectively, it is also urgently needed to examine comprehensively systematic comparative studies to validate the performances in multiple methods and data sets in real data application and simulation studies for longitudinal time series data or its meta data in the field.

## 4. Online Methods

### 4.1. Batch-Correction Methods

To explore the influence of batch factors on the detection of group differences in longitudinal trajectory tests, we employed the existing state-of-the-art batch correction methods that have been widely adopted for high-throughput data. i. ARSyNSeq [25] (previously ARSyN and ASCA): ARSyNSeq is based on two stepwise filtering criteria by using ANOVA simultaneous component analysis (ASCA), which was initially developed by Smilde et al. [41] as an exploratory analysis tool. The total variation of given data is partitioned into sub-models of biological factors such as the main factors, time, treatment group, and its interaction term explained by principal components to find the residuals of such signals, which has been referred to as noise of signals in the study. Next, the noise of signals is further used to identify the structured noise of errors, which is referred to as signals of errors and batch sources. The filtered data by both noise of signals and signal of errors can be utilized for the inferential analyses of temporally differential expression. ii. ComBatSeq (previously Combat) [24]: On the basis of the empirical Bayes approach, we borrow the sharing information across genes for the given batch factor, which is more robust when having the small sample size. The filtered data by the batch estimates are given for the subsequent downstream analyses. Recently, the empirical Bayes approach has been further implemented for count data as ComBatSeq in the sva package [42]. iii. Harman [21]: It is based on the metrics of the extent to which data contamination due to batch factors exists, and the raw variation is preserved on the batch-corrected data by using the *p* values computed from gPCA [23] (guided-PCA) in the previous literature. After independently removing the batch sources on each of the principal components, the modified matrices are transformed to the original batch-free data, which can be directly applied for the subsequent downstream analyses.

### 4.2. Group Difference Detection Methods for Longitudinal Trajectory Tests

For the purpose of identifying time-varying group differences, we employed the following set of statistical methods that have been developed for factorial time course experimental designs in longitudinal settings. i. splinectomeR [43]: It has been developed for longitudinally measured microbiome time series data for the purpose of detecting taxon abundance and α diversity between groups over time. The method is performed by the underlying mechanisms of loess spline fitting procedures and permutated statistical significance determination if the observed difference with enrichment is truly confident compared to shuffled null distribution. ii. microbiomeDASim (incorporated w/metaSplines) [30,44]: It is publicly available software to enable the generation of mimic microbiome data to fully account for several features, such as highly unmeasured zero reads, nonnegative values, and correlated structure within the subjects in repeated measurements, which inherently exist in current microbiome longitudinal data at taxonomy levels. It is also facilitated using metaSplines, which was proposed in the original study of the metagenomeSeq R package [30,44] and named Gaussian smoothing-spline ANOVA. iii. metamicrobiomeR [34]: It has been proposed by generalized additive models with location, scale, and shape parameters to allow the adjustment of confounded factors, and the estimated coefficients represent the log (odds ratio) comparing the target treatment group versus the control group over time in longitudinal settings. iv. dream (incorporated w/voom) [39,45,46]: We also employed the dream approach incorporated with the voom package that has been developed for more complicated experimental designs in RNA-Seq count data in our study.

### 4.3. Pre-Processing Bioinformatics Procedures

The quantified taxonomy abundance table with raw counts for 1392 OTUs by 90 samples over time was generated with greengene annotation (Version No. most recent 13_8) and the closed reference clustering by the guidance of tutorials (https://docs.qiime2.org/2021.11/tutorials/qiime2-for-experienced-microbiome-researchers/, accessed on 11 November 2020) of experienced microbiome researchers on the QIIME 2 (Version No. 2020.11) bioinformatics tool method [47]. After trimming, the filtered data with at least 10 nonzero values for samples, resulting in 743 OTUs, were further used for the subsequent analyses to examine batch impact on downstream analyses.

### 4.4. Experimental Design for Longitudinal Microbiome Time Series Data with Two Different Treatment Groups

Rumen liquid samples were collected via the rumen cannula from 10 Holstein cows at mid-lactation. Longitudinal and repeated samples were collected 9 times over the 55-day period. The samples were collected at 8:00 to 8:30 am on each sampling day, snap-frozen in liquid nitrogen and then stored at −80 °C. Total DNA was extracted using a QIAamp PowerFecal DNA Kit (Qiagen, Germantown, MD, USA). DNA integrity and concentration will be quantified using a BioAnalyzer 2100 (Agilent, Palo Alto, CA, USA). The 16S rRNA gene sequencing was performed as we previously described [48].

## Figures and Tables

**Figure 1 genes-13-00392-f001:**
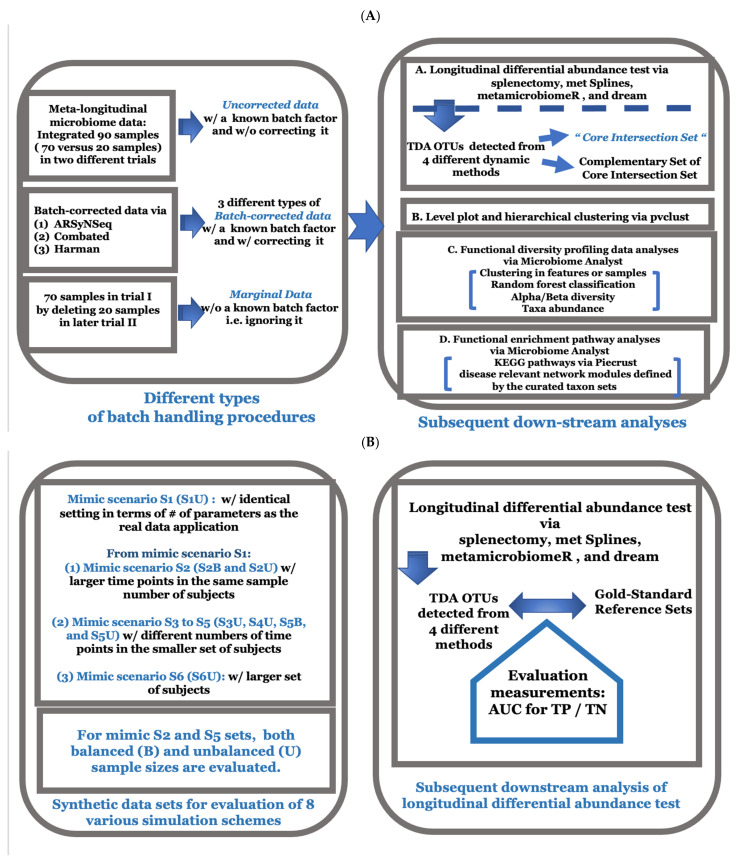
(**A**) Depicting a schematic illustration of this entire study in real data application. (**B**) This figure describes various simulation schemes in this study.

**Figure 2 genes-13-00392-f002:**
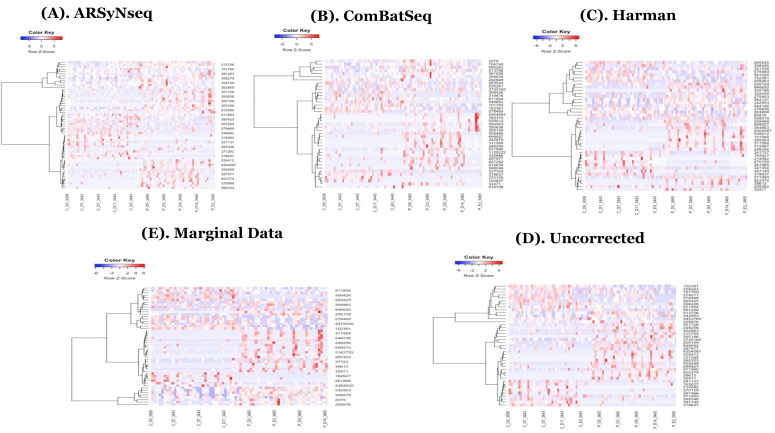
(**A**–**E**) show the heatmaps generated by the core-intersection sets from 5 different types of batch handling procedures (ARSyNseq, ComBatSeq, Harman, Uncorrected, and Marginal Data) in clockwise from top-left.

**Figure 3 genes-13-00392-f003:**
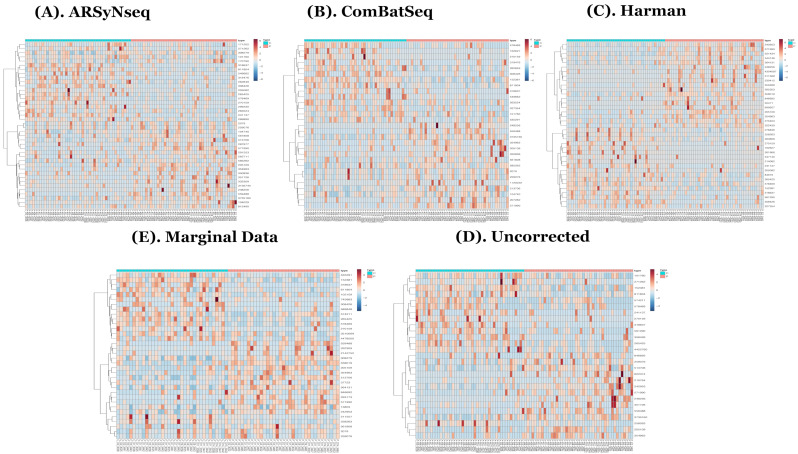
(**A**–**E**) show heatmaps generated from the MicrobiomeAnalyst [27,28] at taxonomy feature levels from core intersection sets detected by multiple longitudinal differential abundance test methods in 5 different types of batch handling procedures (ARSyNseq, ComBatSeq, Harman, Uncorrected, and Marginal Data in clockwise from top-left) after filtering out low abundant features via default criteria in MicrobiomeAnalyst [27,28] from Figure 2A–E. Note that type C and P represent a control and target treatment group, respectively.

**Figure 4 genes-13-00392-f004:**
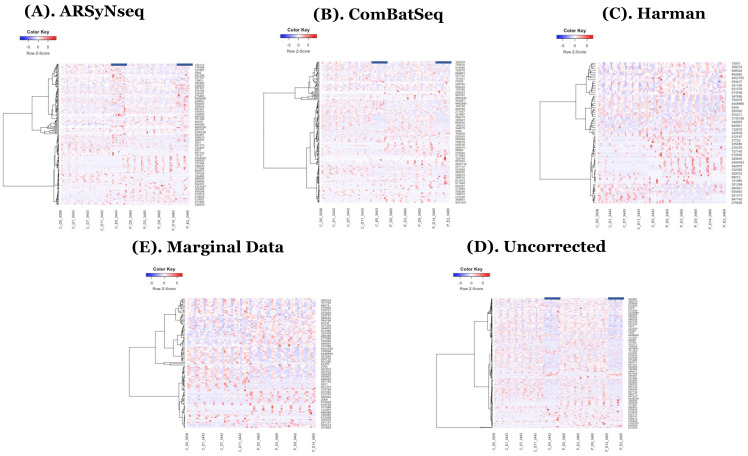
(**A**–**E**) show the heatmaps generated by the complementary sets of core-intersection sets from 5 different types of batch handling procedures (ARSyNseq, ComBatSeq, Harman, Uncorrected, and Marginal Data) in clockwise from top-left. Note that blue bars indicate samples that still show batch effect in two corrected datasets (ARSyNSeq and ComBatSeq) and uncorrected data.

**Figure 5 genes-13-00392-f005:**
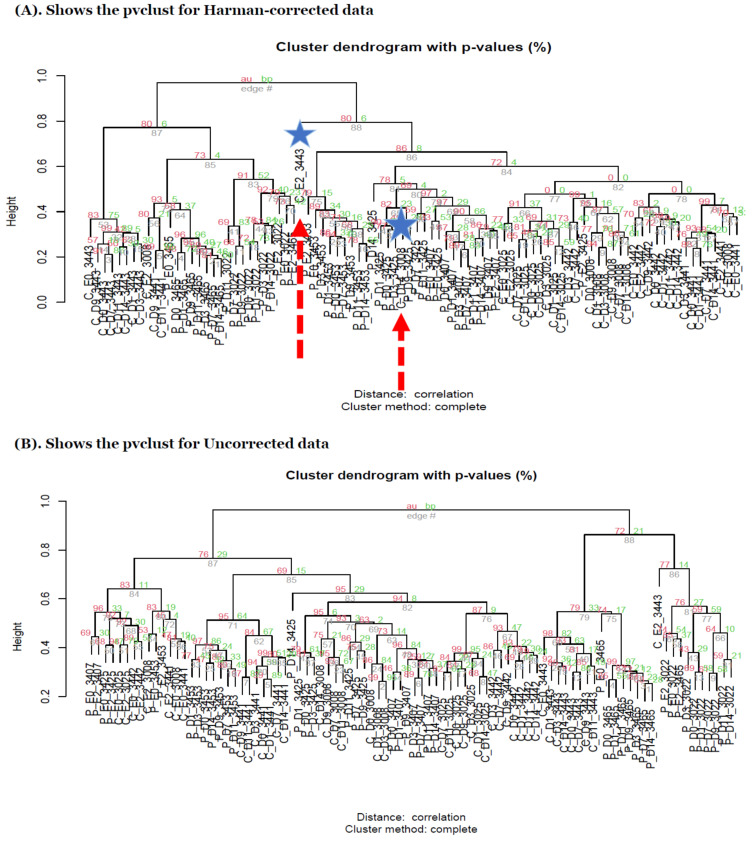
(**A**) shows the pvclust for Harman-corrected data. AU *p*-value in red color represents the area calculated by multi-scale bootstrap resampling procedure, whereas BP value in green color represents the bootstrap probability. In general, BP value is less accurate than AU value and AU value is recommended to be selected in the package currently [26]. Note that Harman-corrected data present much tighter grouping clusters in intra-samples except in 3008 at D-14 and 3443 at E-2 subject (highlighted in stars) in control group on pvclust with complete [26], compared to Uncorrected data with more mixed-up patterns between two treatment groups to cluster samples in (**B**). (**B**) shows the pvclust for uncorrected data. AU *p*-value in red color represents the area calculated by multi-scale bootstrap resampling procedure, whereas BP value in green color represents the bootstrap probability. In general, BP value is less accurate than AU value and AU value is recommended to be selected in the package currently [26]. (**C**–**F**) show the dendrograms generated from MicrobiomeAnalyst [27,28] for Harman and uncorrected data in left panel and the abundance of taxonomy genus levels in right panel. The type C (red) and P (green) in (**C**,**D**) represent control and P drug target treatment group, respectively [27,28]. Note that the sample clustering in plot tree dendrograms is conducted by Ward method in MicrobiomeAnalyst [27,28]. (**G**–**J**) shows the β diversity profiling 2D-PCoA and classification performances for random forest generated from MicrobiomeAnalyst [27,28] in Harman (left panel) and uncorrected data (right panel), respectively. Type C and P represent the control and drug target treatment group, respectively. In β diversity profiling analyses, distance method is used by Bray–Curtis Index and statistical significance test is used by permutational MANOVA (PERMANOVA) [27,28].

**Figure 6 genes-13-00392-f006:**
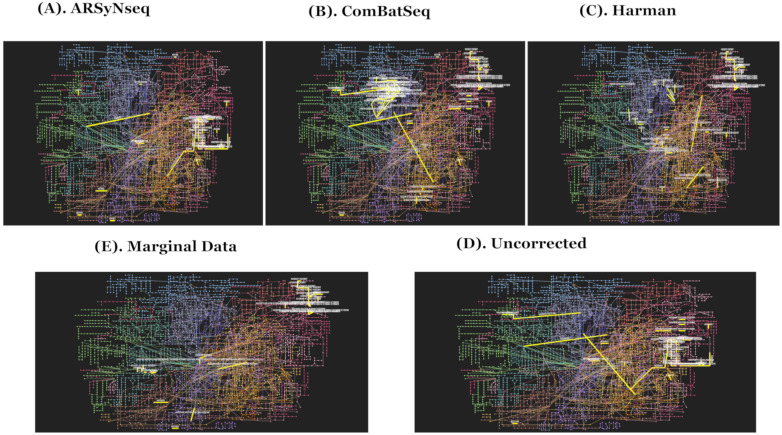
(**A**–**E**) show the network enrichment KEGG pathways generated from MicrobiomeAnalyst [27,28] to represent overall functional profiling in association analyses (ARSyNseq, ComBatSeq, Harman, Uncorrected, and Marginal Data) in clockwise from top-left. Note that top 5 significant modules in yellow edges are presented in figures [27,28].

**Figure 7 genes-13-00392-f007:**
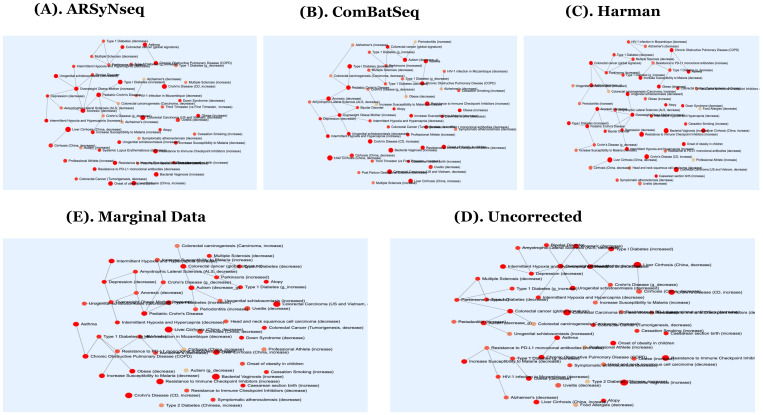
(**A**–**E**) show the network modules generated from MicrobiomeAnalyst [27,28] in TSEA (taxon set enrichment analyses) defined by host intrinsic taxon sets (ARSyNseq, ComBatSeq, Harman, Uncorrected, and Marginal Data) in clockwise from top-left [27,28].

**Figure 8 genes-13-00392-f008:**
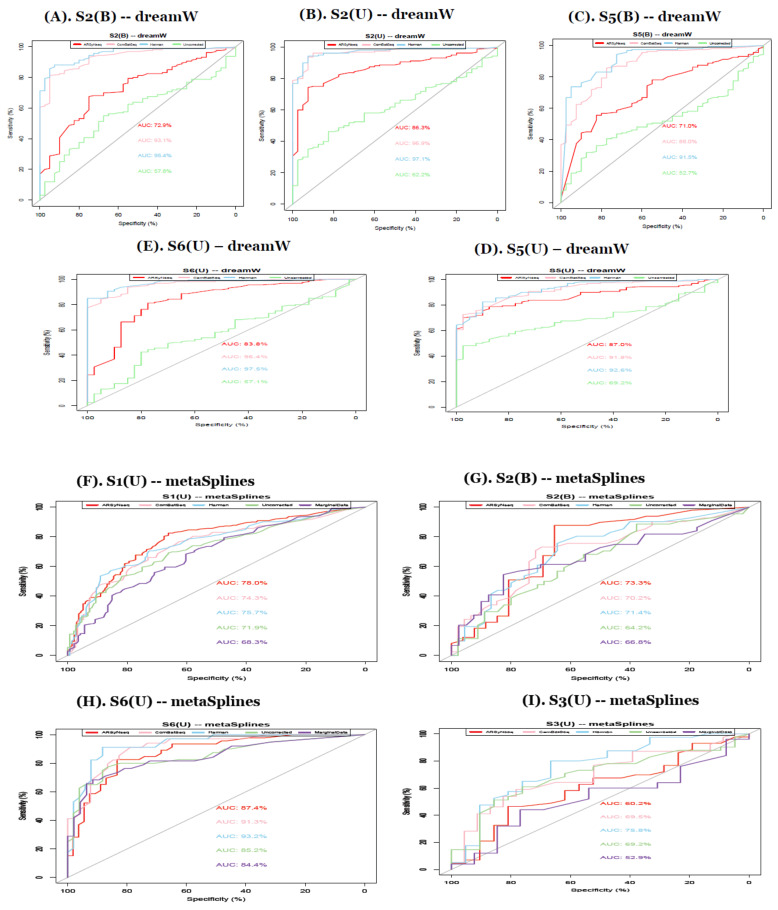
(**A**–**E**) show the ROC curves in simulation studies to confirm that Harman-corrected data show the best performance as shown in real data application. (S2B, S2U, S5B, S5U, and S6U for dream method in clockwise from top-left) dreamW represents the model including primer. Sets of the known batch factor as one of main factors and dream WO excludes it in the model. (**F**–**I**) show the ROC curves in simulation studies to confirm that Harman-corrected data show the best performance as shown in real data application. (S1U, S2B, S3U, and S6U for metaSplines method in clockwise from top-left) dreamW represents the model including primer. Sets of the known batch factor as one of the main factors and dream WO excludes it in the model.

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
