# Peer review of "Large-Scale Meta-Longitudinal Microbiome Data with a Known Batch Factor"

_genes, 2022, doi:10.3390/genes13030392_

Round 1

Reviewer 1 Report

Minor comments:

L29: “Ageing” is not a disease. You can use “ageing-related diseases” instead.

L62: How many samples in the first trial?

L62-64: Please indicate the statistical test you used for this calculation. What is this delta value? How is it calculated? Should also be explained in the results.

L126-128: It is better to have the results for the batch effect on functional enrichment (PICRUSt results) as a summary figure in the  main figures, in my opinion. Although Harman performed relatively worse, it will be a fair comparison for the reader.

L169: This information should go into the relevant figure caption.

L291-296: Please describe the pre-processing of the 16S data in more detail. How the filtering was done, which clustering method was used? Considering regular updates in QIIME 2 version, please indicate the version used.

Comments on the Figures:

Figure 2 and 3) Please add a title for each sub-figure showing which is from which method. Not sure if Harman is explicitly performing better than the other tools here.

Figure 3A-E) What are the blue bars on the heatmaps?

Figure 4A-B) You need to explain in the figure caption what do “stars” denote and the meaning of green/red numbers.

Figure 4G) Please add titles (e.g. Harman-corrected & Harman-uncorrected) for each sub-figure. You need to explain what are the different colors (what is “C” and “P”) in the figure caption. You also should mention the method used for measuring the beta diversity in the figure caption. What is the variation explained by “type” in the corrected & uncorrected beta diversity plot? (to be calculated by PERMANOVA for example). Please include this result in the PCoA plots.  If you make the same comparison for the other correction methods, how is it compared to Harman?

General comments:

Figure texts are too small to read. Figure legends should be crystal clear to the reader and captions need to be more informative. Please add 1-2 sentences into the Discussion about the performance of Harman in functional predictions.

Author Response

" Please see the attachment "

Reviewer 2 Report

The authors present study of the influence of batch effect on certain types of analyses (differential abundance, functional and enrichment analyses) and a comparison of different batch effect mitigation techniques on the observed batch effects. With growing number of microbiome studies, I believe that looking at the effect of batch and comparing batch effect mitigation techniques is something that needs to be undertaken, however, the current study has major issues that does not allow it to be reviewed adequately in its current state.

The largest issue is that the manuscript is not written cohesively and in a way that allows the reader to fully understand and interpret the results. Results in the result section are given without any context to what data is being analyzed, what kind of data is being analyzed and where it was derived from, what tests are being performed and why, and how the results of the tests being performed relate to one another. Certain terms such as “meta-longitudinal”, “TDA”, “core intersection set”, “complementary set”, etc. are given, but never given a definition or explained. What appears to be sample names are referred to, but were not described or introduced in a way that allows the reader to know its relevance. The schematics that are supposed to provide an easy, convenient way to understand the study design are more confusing than should be, and contain the same terms that have not been defined for the reader. Figures and figure legends have far too little detail about what each show, which makes it difficult to interpret and know what each is showing. All the above mentioned issues add up to make one unable to adequately review the current manuscript. There is also much detail missing from the methods section (where was the dataset used for comparison study downloaded from? How were certain statistics mentioned in results but not methods performed (e.g. PCoA, random forest)? How was the real data collected treated and analyzed? How was hierarchical clustering performed? These are only some of what is missing.).

Additionally:

  • it is also odd that none of the more popular differential abundant testing methods were included in this study. Popular methods such as DESeq2, metagenomeSeq, ANCOM, edgeR, mixed models, etc. can all be used to analyze longitudinal data. Including these types of differential abundance methods would make the manuscript much more relevant to the broader microbiome research community.
  • There is a newer method, MMUPHin, that has been developed for reducing batch effects in meta-analysis, which is not included here, but should be.
  • “Movies” were referred to in the manuscript, but I did not see any video files included with the manuscript.

Author Response

" Please see the attachment "

Round 2

Reviewer 1 Report

My major concern about this study is that the analysis is still very messy and I am not convinced that Harman is not explicitly performing better as stated here. Furthermore, figure captions still do not include enough information and figures & text is not well-structured. Please see some comments below:

L350: "otu_ids" does not tell anything. Please rather use "OTU". Still do not include sufficient information. The reader should be able to repeat the same analysis, if they need to. What is the clustering method used?  Which tutorial in QIIME2 was used? etc etc. Also, the language used here is not very clear.

Figure 3: Blue bars should be explained in the figure caption.

Figure 2-3: The statement about the "better performance" of the Harman in these figures still lacks a statistical base.

Figure 4A-B) "You need to explain in the figure caption what do “stars” denote and the meaning of green/red numbers" ---> As I recommended before, these info should be in the figure caption, not in the main text. The reader can not go into the main text each time, to understand the meaning of a symbol, bar, color etc in a figure. 

Figure 5: The figure is very incomprehensible..Very messy, hard to read. The sub-figures are even not denoted by a letter (A, B, C etc).

L168-171: Please correct for the language.

Usage of "5 different datasets" throughout the text is quite confusing. It is not 5 different datasets analyzed here. but it is the same dataset where 5 diff methods are used for correction (or not corrected).

"Please add 1-2 sentences into the Discussion about the performance of Harman in functional predictions." --> This is dwelled on now in the results, but not really discussed in the discussion.

Reviewer 2 Report

Please place the reasoning for not including other popular differential abundance methods in the discussion of the manuscript, as readers who are familiar with differential abundance analysis will be wondering why these are not included.
